# Bridging Vision, Language, and Brain: Whole-Brain Interpretation of Visual Representations via Information Bottleneck Attribution

## Abstract

Understanding how the human brain processes and integrates visual and linguistic information is a long-standing challenge in both cognitive neuroscience and artificial intelligence. In this work, we present two contributions toward attributing visual representations in the cortex by bridging brain activity with natural modalities. We first align fMRI signals with image and text embeddings from a pre-trained CLIP model by proposing a whole-brain representation module that follows anatomical alignment, preserves voxel spatial topology, and captures distributed brain dynamics. Building on this foundation, we further develop an Information Bottleneck-based Brain Attribution (IB-BA) method, which extends information-theoretic attribution to a tri-modal setting. IB-BA identifies the most informative subset of voxels for visual tasks by maximizing mutual information with image and text embeddings while enforcing compression relative to perturbed brain features. Experiments demonstrate superior cross-modal retrieval performance and yield more interpretable cortical attribution maps compared to existing approaches. Collectively, our findings point to new directions for linking neural activity with multimodal representations.

## 1 Introduction

A fundamental question in cognitive neuroscience and artificial intelligence concerns the manner in which the human brain integrates visual and linguistic information (Huth et al., 2016; Fedorenko & Thompson-Schill, 2014). Recent advances in multimodal representation learning, exemplified by CLIP, have demonstrated powerful alignment between images and text (Radford et al., 2021). However, the neural mechanisms underlying comparable cross-modal integration in the brain remain elusive (Kriegeskorte & Douglas, 2018; Schrimpf et al., 2021). Addressing this gap is imperative for enhancing our knowledge of human cognition and for cultivating brain-inspired and interpretable AI systems (Yamins & DiCarlo, 2016; Hassabis et al., 2017).

In computational cognitive neuroscience, deep learning has become central to predicting brain responses to sensory stimuli, a paradigm known as fMRI encoding (Naselaris et al., 2011). These models have advanced our understanding of how sensory features map onto voxel activations, yet they typically emphasize isolated voxels rather than distributed patterns that are essential for cognition (Haxby et al., 2001; Wu et al., 2020). Parallel progress in fMRI decoding has moved from early work on coarse object categorization (Cox & Savoy, 2003; Kay et al., 2008; Walther et al., 2011; Zhou et al., 2024) to recent breakthroughs in reconstructing and retrieving natural images (Lin et al., 2022; Takagi & Nishimoto, 2023; Ozcelik & VanRullen, 2023; Scotti et al., 2023; 2024; Li et al., 2025) and associated text (Ferrante et al., 2023; Ren et al., 2024; Xia et al., 2024; Shen et al., 2024; Qiu et al., 2025). Despite these successes, decoding has been criticized as "wishful thinking" (Vigotsky et al., 2024), since it prioritizes performance over probing the mechanisms of neural representations. Specifically, models operate on incoherent voxel vectors with subject-specific patterns that are highly responsive to visual tasks (Takagi & Nishimoto, 2023; Ozcelik & VanRullen, 2023; Scotti et al., 2023; 2024; Gong et al., 2024b), while ignoring inter-voxel interactions (Wu et al., 2020) and functional connectivity across brain regions (Fingelkurts et al., 2005; Park & Friston, 2013). Moreover, current dense networks fail to preserve the brain's intrinsic spatial topology, further hindering interpretability and reliable backward attribution.

In parallel, existing attribution methods (Selvaraju et al., 2017; Petsiuk et al., 2018; Chefer et al., 2021) are not well-suited for investigating the mechanisms of brain representations. They typically depend on explicit ground-truth labels to assess feature relevance, which are generally unavailable for neural data. Recent advances in multimodal interpretability have largely focused on vision-language models (Radford et al., 2021), where attribution is confined to image-text pairs (Wang et al., 2023). By contrast, interpreting brain activity requires a triadic perspective that bridges voxel-level patterns with both visual semantics (vision) and conceptual (linguistic) representations.

*Can we bridge brain activity with natural modalities to advance our understanding of human cognition?* We approach it from two complementary perspectives: modeling whole-brain representations and developing interpretable brain attribution methods. First, we propose a whole-brain representation module that aligns brain activity with visual and linguistic representations via contrastive learning. The module learns brain embeddings that are aligned with image and text embeddings from a pretrained CLIP model. It respects the anatomical alignment established during fMRI preprocessing, incorporates 3D patch embeddings to capture the spatial topology of voxel activity, and employs self-attention to model distributed brain dynamics. We evaluate the module through bidirectional cross-modal retrieval (Brain-Image/Text and Image/Text-Brain) on the Natural Scenes Dataset (NSD), achieving superior performance compared to state-of-the-art brain decoding models.

Building on this alignment, we introduce a brain attribution method, termed Information Bottleneck-based Brain Attribution (IB-BA), to interpret the relationships between conceptual and visual representations and brain activity. Inspired by Information Bottleneck Attribution (IBA) (Schulz et al., 2020) and its multimodal extension M2IB (Wang et al., 2023), IB-BA extends the framework to a tri-modal setting encompassing brain, image, and text. The method perturbs intermediate feature layers of the brain encoder and identifies voxels that are most informative for visual tasks by maximizing mutual information with image and text embeddings while minimizing redundancy with perturbed features. Experimental results show that IB-BA outperforms commonly used perturbation-based (Petsiuk et al., 2018), gradient-based (Selvaraju et al., 2017), and attention-based (Chefer et al., 2021) methods and enables the exploration of visual representation mechanisms in the brain cortex.

In this work, we make the following contributions:

- We align fMRI activity with visual and linguistic embeddings through CLIP-style contrastive learning by proposing a whole-brain representation module that preserves spatial topology and captures distributed brain dynamics.
- We develop the Information Bottleneck-based Brain Attribution (IB-BA), extending information-theoretic attribution to a three-modality setting (brain, image, text) to identify informative voxel subsets for mechanistic studies in neuroscience.
- We evaluate the proposed methods on the NSD dataset, demonstrating the state-of-the-art cross-modal retrieval performance and more interpretable cortical attribution maps.

## 2 RELATED WORKS

**FMRI Decoding.** FMRI decoding aims to recover human perceptual states across a range of tasks, from coarse-grained object category recognition (Kay et al., 2008; Walther et al., 2011; Zhou et al., 2024) to fine-grained cross-modal retrieval (Lin et al., 2022; Scotti et al., 2023; 2024; Li et al., 2025) and reconstruction of natural images (Takagi & Nishimoto, 2023; Ozcelik & VanRullen, 2023; Scotti et al., 2023; 2024; Li et al., 2025) or textual descriptions (Ferrante et al., 2023; Ren et al., 2024; Xia et al., 2024; Shen et al., 2024; Qiu et al., 2025). Recent studies have further extended decoding to video (Chen et al., 2023; Gong et al., 2024a; Lu et al., 2024), audio (Liu et al., 2024; Denk et al., 2023), 3D pictures (Gao et al., 2024), and language (Ye et al., 2025). However, these methods primarily focus on the pursuit of accurate results while overlooking the underlying neural mechanisms.

**Brain Representation.** Previous work has modeled brain activity by selecting task-relevant voxel vectors and applying linear ridge regression (Takagi & Nishimoto, 2023; Ozcelik & VanRullen, 2023), nonlinear MLPs (Scotti et al., 2023; 2024), or customized Fourier models (Gong et al., 2024b). In medical imaging, 3D MRI volumes are often decomposed into 2D slices and processed using CNNs or ViTs (Kang et al., 2021; Alp et al., 2024), while GNNs have been employed to operate on ROI-level functional connectivity graphs (Li et al., 2021; Zheng et al., 2024a;b). However,

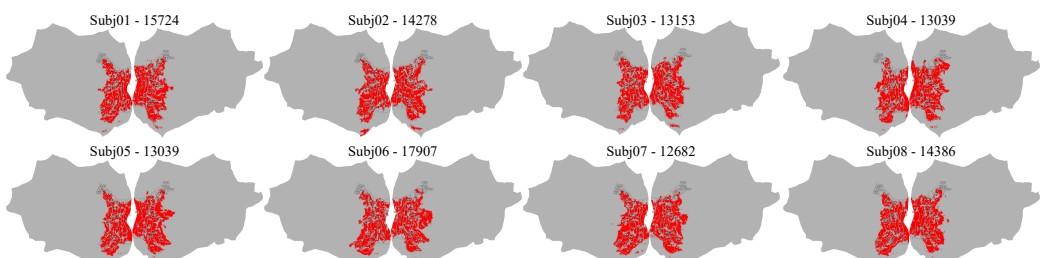

Figure 1: The voxels with high response to visual stimulation in 8 subjects.

these approaches are limited in capturing the full spatial topology of whole-brain voxel activity. For whole-brain modeling, 3D CNNs have been applied in both disease diagnosis (Kim et al., 2020) and functional decoding (Kong et al., 2025). Shen et al. (2024) introduced 3D patch embeddings to represent brain tokens and employed self-attention to capture long-range dependencies, similar to UNETR (Hatamizadeh et al., 2022), although their tokens are derived only from task-relevant regions rather than the full brain.

**Model Attribution.** Numerous methods have been proposed to improve the interpretability of deep neural networks, which can be broadly categorized into three classes. First, perturbation-based approaches, such as RISE (Petsiuk et al., 2018), estimate the contribution of input features by systematically masking or perturbing them and observing the resulting effect on model outputs. Second, gradient-based Grad-CAM (Selvaraju et al., 2017), leverage the gradients of the output with respect to intermediate feature maps to identify salient regions. Third, attention-based methods, such as Chefer et al. (Chefer et al., 2021), utilize the internal attention weights of transformer-based models to highlight important features. More recently, Information Bottleneck Attribution (IBA) (Schulz et al., 2020) has been proposed to identify informative components in feature representations, and its multimodal extension M2IB (Wang et al., 2023) provides explanations for image-text alignment. In this work, we adapt this information-theoretic attribution to brain modeling and further extend it to a tri-modal setting encompassing brain, image, and text.

## 3 PRELIMINARIES

**Natural Scenes Dataset.** We leverage the Natural Scenes Dataset (NSD) (Allen et al., 2022), a large-scale fMRI dataset in which participants viewed 73,000 richly annotated natural images from the COCO dataset (Lin et al., 2014). Each subject was presented with up to 10,000 distinct images across multiple sessions, while high-resolution whole-brain responses were recorded. NSD provides both the scale necessary to train deep models and fine-grained voxel-level coverage for investigating distributed brain representations.

**Brain Region Analysis.** Previous studies often selected non-contiguous, stimulus-responsive voxels, providing only a partial view of brain representations. For each subject, approximately 15,000 of the most strongly activated voxels are selected in the visual cortex, but their coverage is incomplete and highly variable across individuals (Fig. 1).

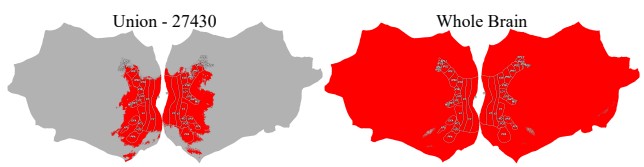

Figure 2: Visual cortex (collection of high-response voxels in 8 subjects) and whole-brain cortex.

Combining voxels across eight subjects yields over 27,000 voxels, covering most of the visual cortex but still omitting higher-order regions (Fig. 2, left). In contrast, our whole-brain model incorporates all voxels, enabling the identification of visual representation patterns both within visual areas and across non-visual regions, providing a more comprehensive account of brain activity (Fig. 2 right).

## 4 METHODS

To investigate visual representations in the brain, we first align brain activity with visual and linguistic modalities within a shared representational space. Building on this alignment, we introduce

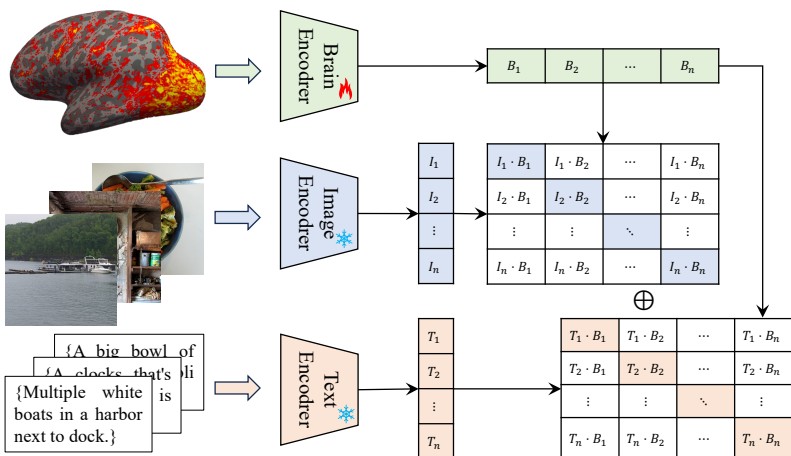

Figure 3: Bridging brain activity with visual-language through contrastive learning.

Information Bottleneck-based Brain Attribution (IB-BA), which extends information-theoretic attribution to the tri-modal brain-image-text setting and identifies voxel subsets that are most informative for visual tasks.

## 4.1 BRAIN VISUAL-LANGUAGE MODEL

To establish alignment between the brain and visual/text modalities, we build upon a pretrained CLIP model and design a brain encoder that maps fMRI voxels into the shared embedding space. Our whole-brain representation module processes 3D fMRI volumes, enabling the capture of distributed and comprehensive brain representation patterns. We employ contrastive learning to align brain embeddings with image and text embeddings. The overall architecture is illustrated in Fig. 3.

**Whole-Brain Representation.** To capture comprehensive brain activity patterns, we design a whole-brain encoder that preserves both the spatial topology of voxels and brain dynamic patterns. We first normalize fMRI volumes to the standard *MNI152-2mm* space provided by the preprocessing pipeline to account for anatomical variability. The normalized 3D volume is then partitioned into non-overlapping patches, each of which is flattened and linear projected into a latent embedding of dimension $d$. We select valid patches based on a brain mask and discard those non-brain regions. A learnable class token aggregates information from all patches to form the global brain representation, while learnable positional embeddings maintain spatial structure. This sequence is processed by stacked transformer encoder layers (Vaswani et al., 2017) with multi-head self-attention, enabling the model to capture long-range dependencies and distributed patterns across the whole brain. Finally, the class token is projected to obtain the final brain embedding, analogous to the global embedding used in the CLIP image encoder.

**Brain Visual-Language Alignment.** To align brain embeddings with image and text embeddings, we employ a contrastive learning objective similar to CLIP (Radford et al., 2021). Given a batch of $N$ image-text pairs and their corresponding brain activity, we obtain the image embeddings $E_I \in \mathbb{R}^{N \times d}$ and text embeddings $E_T \in \mathbb{R}^{N \times d}$ from the pretrained CLIP encoders, where $d$ denotes the embedding dimensionality. The brain embeddings $E_B \in \mathbb{R}^{N \times d}$ are obtained from the whole-brain encoder. We then compute the contrastive loss between brain-image and brain-text embeddings as

$$
\mathcal{L}_{BI} = -\frac{1}{N} \sum_{i=1}^{N} \left[ \log \frac{\exp\left(E_B^i \cdot E_I^{i\top}/\tau\right)}{\sum_{j=1}^{N} \exp\left(E_B^i \cdot E_I^{j\top}/\tau\right)} + \log \frac{\exp\left(E_I^i \cdot E_B^{i\top}/\tau\right)}{\sum_{j=1}^{N} \exp\left(E_I^i \cdot E_B^{j\top}/\tau\right)} \right],
$$

$$
\mathcal{L}_{BT} = -\frac{1}{N} \sum_{i=1}^{N} \left[ \log \frac{\exp\left(E_B^i \cdot E_T^{i\top}/\tau\right)}{\sum_{j=1}^{N} \exp\left(E_B^i \cdot E_T^{j\top}/\tau\right)} + \log \frac{\exp\left(E_T^i \cdot E_B^{i\top}/\tau\right)}{\sum_{j=1}^{N} \exp\left(E_T^i \cdot E_B^{j\top}/\tau\right)} \right],
$$

(1)

where $\tau$ is the temperature parameter which is learnable in CLIP but fixed to the original CLIP value in our model. The overall loss $\mathcal{L} = \mathcal{L}_{BI} + \mathcal{L}_{BT}$ encourages the brain embeddings to be close to their corresponding image and text embeddings while being distant from non-corresponding ones.

## 4.2 INFORMATION BOTTLENECK-BASED BRAIN ATTRIBUTION (IB-BA)

In this section, we propose Information Bottleneck-based Brain Attribution (IB-BA) to identify subsets of voxels that are most informative for alignment of visual and language modalities. IB-BA extends the information bottleneck attribution to a tri-modal setting (brain, image, text) and employs a fitting term and a compression term to balance informativeness and redundancy.

**Information Bottleneck.** The information bottleneck (IB) principle (Tishby et al., 2000) provides a framework for extracting task-relevant representations by balancing sufficiency and compression. Formally, given an input $X$ and target $Y$, the objective is to learn a representation $Z$ that preserves information about $Y$ while discarding irrelevant details from $X$:

$$\max_{p(z|x)} I(Z;Y) - \beta I(Z;X), \tag{2}$$

where $I(\cdot; \cdot)$ denotes mutual information and $\beta$ gives a trade-off between fitting and compression.

**Problem Formulation.** We apply the information bottleneck to the pretrained brain encoder, which produces a brain embedding $Z_B$ aligned with image and text embeddings. Instead of retraining the encoder, IB-BA introduces a perturbation module, parameterized by $\theta_B$, at an intermediate feature layer. This module adds noise to the intermediate features, controlled by $\theta_B$, before passing them through the remaining fixed layers, yielding a perturbed brain embedding $Z_B$. The parameters $\theta_B$ are optimized to preserve information relevant for cross-modal alignment while discarding redundant information from the original brain activity. This procedure identifies the voxels that are most informative for visual tasks. Formally, the optimization follows the standard information bottleneck objective:

$$\max_{\theta_B} I(Z_B; E_I, E_T) - \beta I(Z_B; X_B), \tag{3}$$

where $Z_B$ is the perturbed brain embedding, $E_I$ and $E_T$ are the image and text embeddings, and $\beta$ controls the trade-off between fitting and compression.

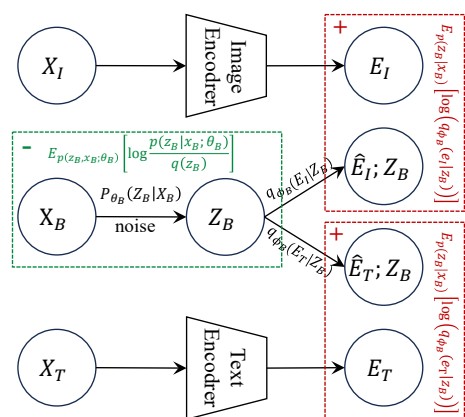

Figure 4: Information bottleneck brain attribution diagram

**Compression term.** The compression term $I(Z_B, X_B; \theta_B)$ measures how much information the brain representation $Z_B$ retains about the input brain activity $X_B$. It can be expressed as

$$I(Z_B, X_B; \theta_B) = D_{\mathrm{KL}}(p(z_B, x_B; \theta_B) \| p(z_B; \theta_B) p(x_B))$$
$$= \mathbb{E}_{p(z_B, x_B; \theta_B)}\left[\log \frac{p(z_B, x_B; \theta_B)}{p(z_B) P(x_B)}\right] = \mathbb{E}_{p(z_B, x_B; \theta_B)}\left[\log \frac{p(z_B | x_B; \theta_B)}{p(z_B)}\right], \tag{4}$$

where $p(Z_B|X_B; \theta_B)$ can be sampled empirically whereas $p(Z_B)$ is intractable. We can use the variational distribution $q(Z_B)$ to approximate $p(Z_B)$, and we have

$$I(Z_B, X_B; \theta_B) = \mathbb{E}_{p(z_B, x_B; \theta_B)}\left[\log \frac{p(z_B | x_B; \theta_B)}{q(z_B)}\right] - \mathbb{E}_{p(z_B, x_B; \theta_B)}\left[\log \frac{p(z_B)}{q(z_B)}\right]$$
$$= \mathbb{E}_{p(z_B, x_B; \theta_B)}\left[\log \frac{p(z_B | x_B; \theta_B)}{q(z_B)}\right] - D_{\mathrm{KL}}(p(z_B) \| q(z_B))$$
$$\leq \mathbb{E}_{p(z_B, x_B; \theta_B)}\left[\log \frac{p(z_B | x_B; \theta_B)}{q(z_B)}\right] \tag{5}$$
$$= \mathbb{E}_{p(x_B)}\left[D_{\mathrm{KL}}(p(z_B | x_B; \theta_B) \| q(z_B))\right],$$

where the inequality is due to the non-negativity of KL divergence. In practice, we set $q(z_B) = \mathcal{N}(z_B; 0, \mathbf{I})$ and $p(z_B | x_B; \theta_B) = \mathcal{N}(\mu_{x_B}, \sigma^2_{x_B}; \theta_B)$, where $\mu_{x_B}$ and $\sigma^2_{x_B}$ are the mean and variance of $Z_B$ over the batch $X_B$.

**Fitting term.** The fitting term $I(Z_B; E_I, E_T; \theta_B)$ measures how much information the brain representation $Z_B$ contains about the image and text embeddings $E_I$ and $E_T$:

$$
\begin{aligned}
I(Z_B; E_I, E_T; \theta_B) &= D_{\mathrm{KL}}(p(z_B; e_I, e_T; \theta_B) \| p(z_B; \theta_B) p(e_I, e_T)) \\
&= \mathbb{E}_{p(z_B; e_I, e_T)} \big[ \log p(e_I, e_T | z_B; \theta_B) \big] - \mathbb{E}_{p(e_I, e_T)} \big[ \log p(e_I, e_T) \big].
\end{aligned}
\tag{6}
$$

The second term is the joint entropy of the variables $E_I$ and $E_T$, which is a constant independent of $\theta_B$ usually ignored in the optimization problem. The conditional probability distribution $p(e_I, e_T | z_B; \theta_B)$ in the first term is not in general tractable. Therefore, we approximate it with a variational distribution $q_{\phi_B}(e_I, e_T | z_B)$, and we have

$$
\begin{aligned}
&I(Z_B; E_I, E_T; \theta_B) \\
&\doteq \mathbb{E}_{p(z_B; e_I, e_T)} \big[ \log q_{\phi_B}(e_I, e_T | z_B; \theta_B) \big] + \mathbb{E}_{p(z_B; e_I, e_T)} \Big[ \log \frac{p(e_I, e_T | z_B; \theta_B)}{q_{\phi_B}(e_I, e_T | z_B; \theta_B)} \Big] \\
&\doteq \mathbb{E}_{p(z_B; e_I, e_T)} \big[ \log q_{\phi_B}(e_I, e_T | z_B; \theta_B) \big] + \mathbb{E}_{p(z_B)} \big[ D_{\mathrm{KL}}(p(e_I, e_T | z_B) \| q(e_I, e_T | z_B)) \big] \\
&\geq \mathbb{E}_{p(z_B; e_I, e_T)} \big[ \log q_{\phi_B}(e_I, e_T | z_B; \theta_B) \big].
\end{aligned}
\tag{7}
$$

**Total objective.** The total optimization objective of the information bottleneck brain attribution is to maximize the weighted sum of the fitting term and the compression term, expressed as

$$
\begin{aligned}
\theta_B^* &= \arg\max_{\theta_B} \ I(Z_B; E_I, E_T; \theta_B) - \beta I(Z_B, X_B; \theta_B) \\
&= \arg\max_{\theta_B} \ \mathbb{E}_{p(z_B; e_I, e_T)} \big[ \log q_{\phi_B}(e_I, e_T | z_B) \big] - \beta \mathbb{E}_{p(x_B)} \big[ D_{\mathrm{KL}}(p(z_B | x_B; \theta_B) \| q(z_B)) \big].
\end{aligned}
\tag{8}
$$

In practice, we use empirical samples of $x_B$, $e_I$, and $e_T$ to approximate the variational optimization objective $\hat{p}(x_B, e_I, e_T) \doteq \frac{1}{N} \sum_{i=1}^{N} \delta(x_B - x_B^i) \delta(e_I - e_I^i) \delta(e_T - e_T^i)$. Since $e_I \sim p(e_I | x_I; \theta_I)$ and $e_T \sim p(e_T | x_T; \theta_T)$ are conditionally independent, the joint distribution can be decomposed as $q_{\phi_B}(e_I, e_T | z_B) = q_{\phi_B}(e_I | z_B) q_{\phi_B}(e_T | z_B)$. In CLIP-style models, the embeddings are projected and normalized into a shared space by a modality-specific mapping function $f_m(\cdot)$. Thus, the log of the Gaussian probability density $\log q_{\phi_B}(e_m | z_B)$ simplifies and is proportional to the cosine similarity between $f_m(e_m)$ and $f_{\phi_B}(z_B)$, giving the optimization objective of the fitting term

$$
\begin{aligned}
&\max_{\theta_B} \mathbb{E}_{p(x_B, e_I, e_T)} \big[ \log q_{\phi_B}(e_I, e_T | z_B; \theta_B) \big] \\
&\doteq \max_{\theta_B} \mathbb{E}_{p(z_B | x_B)} \big[ \log \big( q_{\phi_B}(e_I | z_B) \big) + \log \big( q_{\phi_B}(e_T | z_B) \big) \big] \\
&\propto \max_{\theta_B} \mathbb{E}_{p(z_B | x_B)} \big[ \cos \big( f_I(e_I), f_{\phi_B}(z_B) \big) + \cos \big( f_T(e_T), f_{\phi_B}(z_B) \big) \big].
\end{aligned}
\tag{9}
$$

# 5 EXPERIMENTS

## 5.1 IMPLEMENTATION DETAILS

**Datasets.** We use fMRI data collected from 8 subjects in the NSD dataset, who viewed a total of 73,000 images (each subject viewed 9,000 unique images and 1,000 shared images) with each image presented 3 times across 40 sessions. Each image is accompanied by a corresponding text description, providing rich visual and linguistic information.

● **Brain.** We use *nsd_mapdata.m*[1] to map the GLMdenoised BOLD signals from the `func1mm` to the `MNI152_T1_1mm` space. The input voxel data is resampled to a 2mm isotropic resolution, resulting in a volume shape of $(91, 109, 91)$, and normalized to zero mean and unit variance as model input. We apply the `MNI152_T1_2mm` brain mask to exclude non-brain patches.

● **Image.** We use the original images from the NSD dataset, which are resized to $224 \times 224$ pixels and normalized.

---

[1] https://github.com/cvnlab/nsdcode

Table 1: Brain multimodal retrieval accuracy.

| Methods | Brain-Image | | | Image-Brain | | | Brain-Text | | | Text-Brain | | |
|---|---|---|---|---|---|---|---|---|---|---|---|---|
| | R@1 | R@5 | R@10 | R@1 | R@5 | R@10 | R@1 | R@5 | R@10 | R@1 | R@5 | R@10 |
| Ridge | 22.07 | 51.36 | 65.54 | 26.14 | 56.94 | 70.68 | 18.07 | 46.33 | 61.17 | 20.61 | 50.31 | 65.17 |
| Mindeye | 32.81 | 63.56 | 75.89 | 27.55 | 59.04 | 72.55 | 23.23 | 52.80 | 57.17 | 20.17 | 49.26 | 63.62 |
| Mindeye2 | 23.46 | 51.37 | 64.77 | 23.05 | 51.31 | 65.09 | 17.79 | 43.23 | 57.26 | 17.57 | 43.34 | 57.03 |
| TGBD | 29.28 | 60.82 | 74.38 | 22.81 | 54.09 | 69.72 | 20.91 | 50.18 | 64.79 | 17.70 | 45.33 | 60.25 |
| Ours | **53.16** | **83.24** | **91.00** | **41.34** | **74.04** | **84.36** | **40.73** | **73.73** | **84.75** | **33.75** | **66.70** | **78.27** |

• **Text.** We do not use the COCO captions provided by NSD (Lin et al., 2014) for training, as they are too brief for effective alignment. Instead, we employ a vLLM (4-bit quantized Qwen2.5-VL-32B) to generate detailed descriptions, constraining their length to within 70 words to ensure compatibility with the CLIP text encoder. These descriptions provide richer semantic information, facilitating more effective alignment between brain and text embeddings. For attribution analysis with IB-BA, however, we revert to the original COCO captions.

**Model Implementation.** The whole-brain representation module is constructed by replacing the CLIP image encoder with a 3D patch embedding (default patch size 14). Except for setting the number of transformer blocks to 12, all other hyperparameters follow the CLIP configuration. We use `CLIP-ViT-H/14` as the default vision-language model. Training is performed on 2 NVIDIA GTX 4090 GPUs with the AdamW optimizer and an initial learning rate of $3 \times 10^{-4}$. The batch size is 256 (128 per GPU), with a memory queue that caches the previous 4096 samples. We apply linear warm-up for the first 1% of training steps, followed by cosine annealing to decay the learning rate to $1 \times 10^{-5}$. The model is trained for 150 epochs. For each {brain, image, text} triplet, we insert a parameterized bottleneck module into the brain encoder to perturb intermediate features. The bottleneck is trained using a single sample, with a batch size of 10 for 20 steps, optimized by Adam with a learning rate of 1.

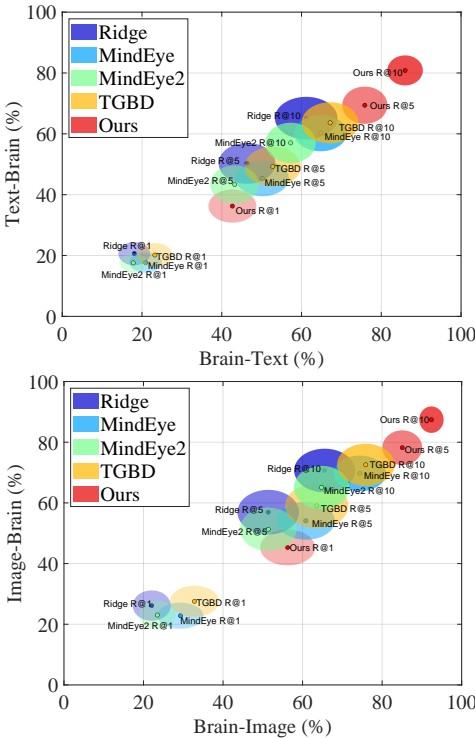

Figure 5: Brain retrieval accuracy.

## 5.2 EVALUATION METRICS

We evaluate the alignment between brain and image/text modalities using cross-modal retrieval tasks. Specifically, we compute the retrieval accuracy (R@1, R@5, R@10) for brain-to-image/text and image/text-to-brain, which measures the proportion of correct matches in the top 1, 5, and 10 retrieval results. For attribution, a major challenge lies in the absence of explicit ground-truth brain maps, and attribution results depend on both the attribution method itself and the underlying model performance. To address this, we adopt degradation-based metrics (Chattopadhay et al., 2018; Wang et al., 2020), which are grounded in the principle that eliminating regions with high attribution scores should lead to a decrease in retrieval performance, whereas eliminating regions with low attribution scores should have little impact or may even improve performance by eliminating irrelevant regions.

## 5.3 RESULTS

**Brain Visual-Language Alignment.** We compare our whole-brain representation module with several neural decoding methods, including Ridge regression (Takagi & Nishimoto, 2023), Mindeye

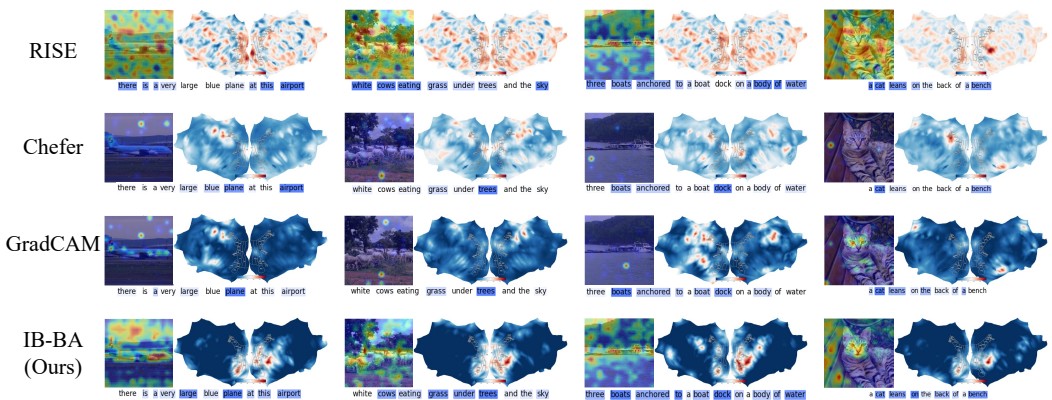

Figure 6: Qualitative results of brain attribution methods.

Table 2: Brain attribution degradation metrics.

| Methods | Conf. Img-Drop↓ | Conf. Img-Incr↑ | Conf. Txt-Drop↓ | Conf. Txt-Incr↑ |
|---------|-----------------|-----------------|-----------------|-----------------|
| RISE | 4.739 | 6.594 | 3.024 | 27.786 |
| GradCAM | 8.435 | 2.480 | 5.859 | 13.932 |
| Chefer | 4.416 | 11.418 | 3.089 | 27.462 |
| IB-BA | **2.201** | **19.339** | **1.666** | **35.549** |

(Scotti et al., 2023), Mindeye2 (Scotti et al., 2024), and TGBD (Kong et al., 2025). As shown in Table 1 and Figure 5, our approach achieves superior performance in both brain-to-image/text and image/text-to-brain retrieval, demonstrating the effectiveness of the whole-brain representation and establishing a reliable basis for attributing brain activity to visual representations. Additional cross-modal retrieval results are provided in Appendix C.1.

**Brain Attribution.** We compare our IB-BA method with commonly used attribution techniques, including RISE (perturbation-based (Petsiuk et al., 2018)), Grad-CAM (gradient-based (Selvaraju et al., 2017)), and Chefer (attention-based (Chefer et al., 2021)). Due to the absence of ground-truth labels for visual representations in the cerebral cortex, directly assessing attribution accuracy is challenging. To address this, we adopt an degradation-based evaluation strategy: brain features are ablated according to the attribution maps, and the resulting changes in cross-modal alignment are analyzed. As shown in Table 2, the IB-BA outperforms these baselines in both degradation metrics and improvement metrics, indicating its effectiveness in identifying informative brain regions for visual tasks. We further evaluate reverse mapping for image attribution maps and word-level attribution. As illustrated in Figure 6, IB-BA produces attribution maps that are more focused and interpretable, effectively highlighting object recognition in images and concept localization in text, while maintaining consistent cross-modal correspondence.Furthermore, we observe that the cortical attribution maps generated by IB-BA align well with known visual cortex, providing stronger biological plausibility than alternative methods. Attribution results for all subjects are provided in Appendix C.2.

## 5.4 ABLATION STUDIES

**Brain Representation.** We conduct ablations on subj01 to examine the effect of modality alignment, brain region selection, and caption sources (Table 3). First, we compare unimodal versus bimodal alignment. Using only image–brain or text–brain pairs fails to yield competitive retrieval performance, whereas combining both modalities produces substantial gains across all retrieval directions, confirming the necessity of tri-modal alignment. Second, we investigate the impact of different brain regions. Using only the visual cortex leads to lower accuracy compared to the whole brain, indicating that non-visual regions also contribute to visual cognition, particularly in supporting the abstract and conceptual dimensions required for textual-alignments. Finally, we evaluate different caption sources. High-quality descriptions from Qwen32B lead to the best results, while

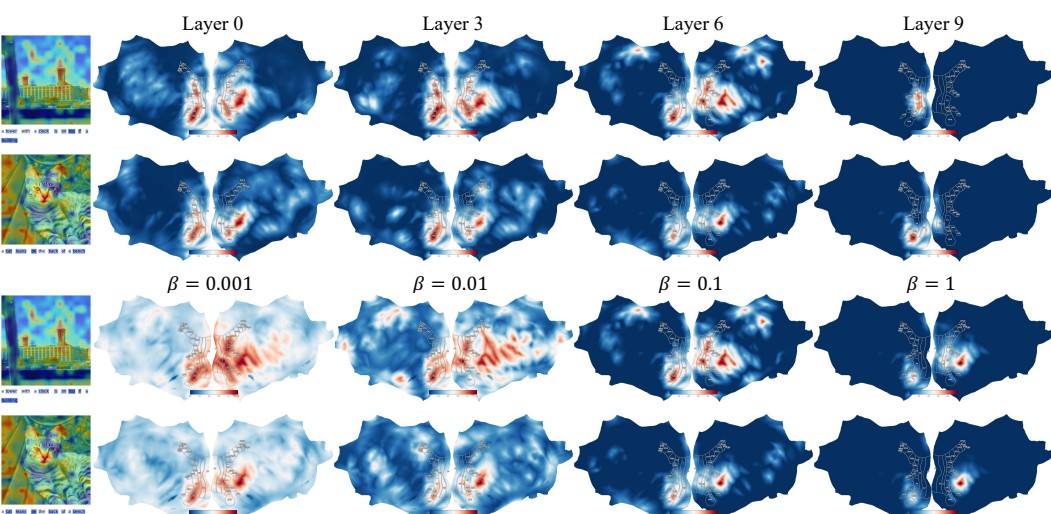

Figure 7: Ablation study of the IB-BA hyperparameter target layer and $\beta$.

Table 3: Retrieval accuracy for the ablation of the module of brain visual-language alignment.

| Modality | | Brain Region | Captions | Top 1 Acc. | | | |
|---|---|---|---|---|---|---|---|
| Image | Text | | | Brain-Image | Image-Brain | Brain-Text | Text-Brain |
| ✓ | ✗ | Whole Brain | Qwen32B | 73.89 | 58.93 | 36.37 | 18.94 |
| ✗ | ✓ | Whole Brain | Qwen32B | 34.14 | 13.97 | 51.83 | 39.81 |
| ✓ | ✓ | Visual Cortex | Qwen32B | 70.30 | 54.50 | 53.71 | 43.27 |
| ✓ | ✓ | Whole Brain | COCO | 60.52 | 44.91 | 32.81 | 29.33 |
| ✓ | ✓ | Whole Brain | llava13B | 68.39 | 54.21 | 44.68 | 36.58 |
| ✓ | ✓ | Whole Brain | Qwen32B | **70.30** | **57.12** | **55.03** | **46.33** |

COCO annotations significantly impair brain-text alignment. This demonstrates that the richness of textual supervision directly impacts the quality of learned brain embeddings.

**IB-BA parameter sensitivity.** We investigate the sensitivity of IB-BA to its hyperparameters (Figure 7). First, we deploy IB-BA at different layers of the brain encoder and observe that spatial extent of the attribution maps decreases progressively from shallow to deeper layers. This reflects the hierarchical aggregation of features across stacked self-attention blocks: shallow layers retain more information due to a permissive bottleneck, whereas deeper layers enforce stricter filtering of salient features. In addition, we examine the effect of different values of $\beta$, which controls the relative weight of the compression term in the information bottleneck. As expected, smaller $\beta$ values yield broader activation maps, while larger $\beta$ values result in more localized maps, highlighting the trade-off between informativeness and compression. We further examine other hyperparameters, including noise variance, learning rate, and the number of training steps, with all quantitative analyses and visualizations reported in Appendix C.2.

## 6 CONCLUSION

In this paper, we bridge brain activity with visual and linguistic modalities by modeling whole-brain representations that capture brain dynamics and preserve spatial topology, enabling attribution of cortical representations beyond models focused solely on decoding results. We further propose IB-BA, an information-theoretic attribution method that leverages the bottleneck's compression property to identify brain regions most informative for cross-modal alignment. Extensive experiments on the NSD dataset demonstrate the effectiveness of our approach in both brain visual-language alignment and brain region attribution, which offers a principled foundation for exploring the human brain.

## STATEMENT OF ETHICS

This research was conducted using the NSD dataset, which is publicly available and was collected with informed consent from all participants. The study protocol was approved by the relevant institutional review boards, ensuring adherence to ethical standards for research involving human subjects. We are committed to maintaining the privacy and confidentiality of the participants' data, and all analyses were performed in accordance with ethical guidelines.

## REPRODUCIBILITY STATEMENT

We provide comprehensive implementation details in the Experiments section, including model architecture, training procedures, and dataset descriptions. We also include ablation studies to analyze the impact of various components and hyperparameters on performance. To facilitate reproducibility, we release the core implementation code in the supplementary materials, enabling researchers to reproduce our results and build upon our framework in future studies.

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

# Appendix

## A   USE OF LLMS

To enhance the readability of the manuscript, large language models (LLMs) were employed for grammar checking and refinement of wording. We confirm that the final version of the manuscript has been carefully reviewed and validated by humans. At no stage were hidden "prompt injections" inserted into the paper.

To enrich the descriptive details available for model training, we incorporate automatically generated captions from vision-language models (LLaVA-v1.6-13B[2] and Qwen2.5-VL-32B[3]). These captions serve as supplementary annotations that complement the original stimuli, thereby providing finer-grained semantic information that can better guide brain-modality alignment. The detailed implementation procedures and representative examples are provided in Section D.

## B   TRAINING TRICK.

Contrastive learning typically benefits from large batch sizes, as they provide a diverse set of negative samples. However, limited GPU memory poses challenges for small-batch training. To address this, we employ several strategies. First, we maintain a dynamically updated queue that caches embeddings from previous batches, effectively increasing the number of negative samples available for contrastive learning. Second, we utilize mixed-precision training, which accelerates computation and reduces memory usage, allowing for larger effective batch sizes within the available GPU memory. Finally, we implement distributed data parallelism (DDP), further enhancing training efficiency and enabling larger batch sizes across multiple GPUs. By combining these techniques, we can effectively train our contrastive learning model despite memory constraints.

## C   RESULTS

### C.1   BRAIN VISUAL-LANGUAGE ALIGNMENT RESULTS

**Subject-wise results.** We provide subject-wise results of brain visual-language alignment in Table 4.

Table 4: Subject-wise brain visual-language retrieval results.

| Methods | Brain-Image | | | Image-Brain | | | Brain-Text | | | Text-Brain | | |
|---------|------|------|------|------|------|------|------|------|------|------|------|------|
| | R@1 | R@5 | R@10 | R@1 | R@5 | R@10 | R@1 | R@5 | R@10 | R@1 | R@5 | R@10 |
| subj01 | 70.30 | 94.68 | 97.98 | 57.12 | 88.63 | 95.42 | 55.03 | 85.86 | 93.54 | 46.33 | 80.91 | 90.29 |
| subj02 | 61.90 | 90.91 | 96.17 | 48.71 | 83.32 | 92.36 | 46.96 | 80.31 | 90.18 | 39.39 | 72.63 | 84.12 |
| subj03 | 47.36 | 78.40 | 88.56 | 37.07 | 71.93 | 83.12 | 33.83 | 69.16 | 81.27 | 29.79 | 62.51 | 75.07 |
| subj04 | 47.36 | 79.97 | 89.47 | 38.83 | 70.97 | 81.94 | 33.33 | 67.49 | 80.33 | 27.20 | 59.79 | 73.89 |
| subj05 | 74.98 | 96.42 | 98.98 | 62.38 | 92.92 | 97.01 | 60.70 | 90.97 | 96.11 | 52.90 | 86.69 | 93.37 |
| subj06 | 61.78 | 89.74 | 95.17 | 51.06 | 84.38 | 91.40 | 46.87 | 82.63 | 91.28 | 41.93 | 76.20 | 86.51 |
| subj07 | 53.59 | 84.10 | 92.44 | 41.51 | 75.96 | 87.77 | 39.89 | 74.37 | 86.27 | 32.92 | 67.62 | 80.36 |
| subj08 | 32.67 | 66.32 | 80.40 | 25.73 | 57.49 | 70.43 | 24.70 | 56.19 | 68.79 | 19.61 | 48.91 | 63.16 |

**UMAP visualization.** We visualize the brain visual-language alignment for each subject using UMAP, as shown in Figure 8. The UMAP plots demonstrate that our method effectively aligns brain representations with corresponding image and text embeddings, forming distinct clusters for each modality. This indicates that the learned brain embeddings capture meaningful semantic information related to both visual and textual stimuli, facilitating cross-modal understanding.

**Representation similarity matrix.** We compute the representation similarity matrix (RSM) between brain, image, and text embeddings for each subject, as shown in Figure 12. The RSMs reveal strong correlations between brain representations and both image and text embeddings.

---

[2]https://huggingface.co/liuhaotian/llava-v1.6-vicuna-13b
[3]https://huggingface.co/Qwen/Qwen2.5-VL-32B-Instruct

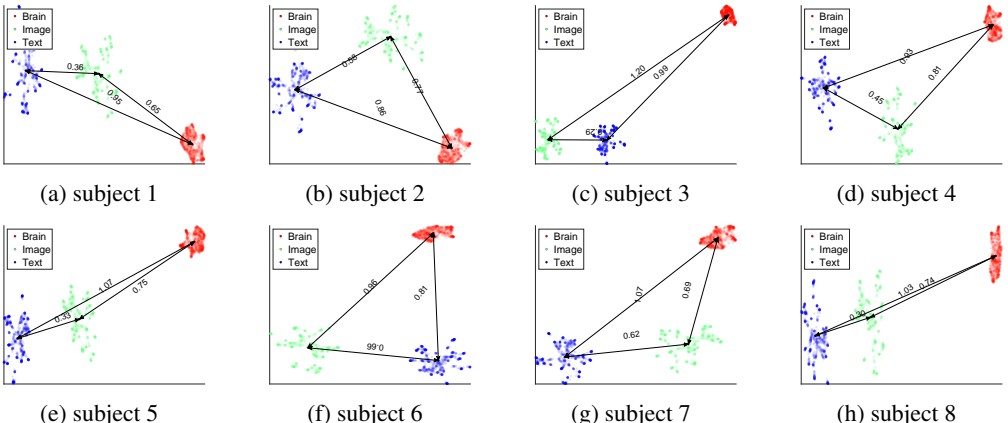

Figure 8: UMAP plot of brain visual-language alignment for each subject.

## C.2 IB-BA ATTRIBUTION RESULTS

**Subject-wise results.** We provide subject-wise results of IB-BA attribution in Table 5. Qualitative results are shown in Fig. 9, where our method highlights brain regions that are more interpretable and relevant to the stimuli.

**Hyperparameter sensitivity analysis.** We conduct a hyperparameter sensitivity analysis of the IB-BA method, varying the target layer, $\beta$, variance of the Gaussian noise, learning rate, and training steps. The results are presented in Table 6 and Figure 10. We find that the filtering effect of the attribution bottleneck intensifies with increasing depth of the self-attention layers, as feature aggregation becomes progressively more hierarchical. Applying IB-BA to shallower target layers, while quantitatively advantageous due to the retention of more information, yields cortical attribution maps that, although concentrated in the visual cortex, fail to differentiate between unique patterns of visual and conceptual representations. Conversely, deeper target layers impose stronger information constraints, producing maps dominated by a single peak. Attribution at intermediate layers provides a balanced solution, retaining sufficient information while distinguishing distinct visual representation patterns. The $\beta$ parameter controls the weight of the compression term in the information bottleneck, thereby regulating the amount of information retained in the attribution. Smaller $\beta$ values preserve more redundant information, while larger $\beta$ values enforce stricter filtering of relevant features. We adopt an intermediate value of $\beta = 0.1$ as a balanced choice. Although quantitative results in Table 6 show that smaller $\beta$ values yield higher scores, this advantage primarily reflects the retention of additional information in ablation evaluations, rather than necessarily indicating superior specificity of the attribution maps. We further evaluated the sensitivity of IB-BA to other hyperparameters, including the noise variance $\sigma$, the learning rate, and the number of training steps. The results indicate that the method is relatively robust to these settings, provided they remain within a reasonable range.

Table 5: Subject-wise brain attribution degradation metrics.

| Methods | Conf. Img-Drop↓ | Conf. Img-Incr↑ | Conf. Txt-Drop↓ | Conf. Txt-Incr↑ |
|---------|-----------------|-----------------|-----------------|-----------------|
| subj01 | 1.748 | 21.7 | 0.958 | 49.9 |
| subj02 | 2.158 | 17.3 | 1.652 | 34.5 |
| subj03 | 3.741 | 11.828 | 2.469 | 29.7845 |
| subj04 | 1.839 | 24.035 | 1.629 | 35.061 |
| subj05 | 2.071 | 18.7 | 1.706 | 33.3 |
| subj06 | 2.825 | 14.516 | 1.977 | 32.688 |
| subj07 | 1.680 | 20.5 | 1.425 | 34.1 |
| subj08 | 1.548 | 26.130 | 1.516 | 35.061 |

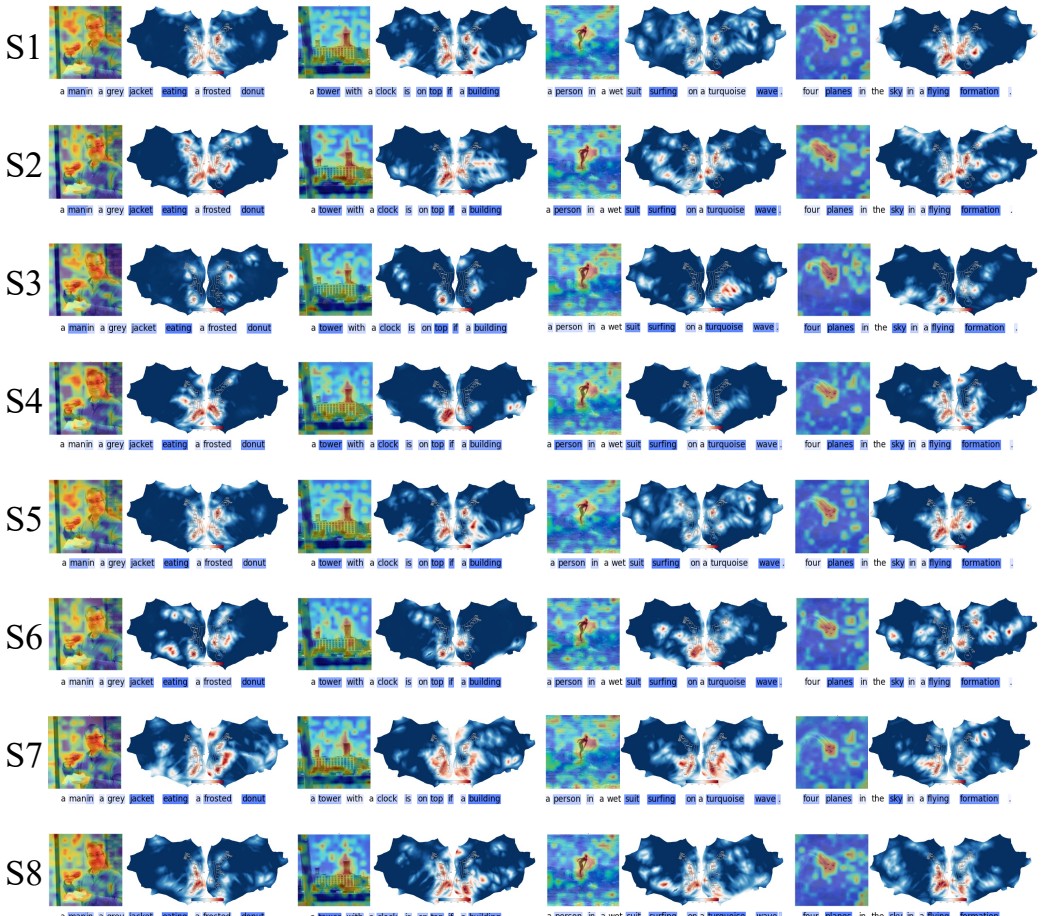

Figure 9: Qualitative results of IB-BA attribution for each subject.

Table 6: Degradation metrics for the hyperparameter ablation study of the IB-BA method.

| | | Param | | | | Conf. | | |
|---|---|---|---|---|---|---|---|---|
| layer | $\beta$ | var | lr | tr steps | Img-Drop↓ | Img-Incr↑ | Txt-Drop↓ | Txt-Incr↑ |
| 0 | 0.1 | 0.1 | 1 | 20 | 0.969037506 | 32.1 | 0.572495244 | 56.4 |
| 3 | 0.1 | 0.1 | 1 | 20 | 1.092111657 | 29.3 | 0.623991036 | 55.3 |
| 6 | 0.1 | 0.1 | 1 | 20 | 1.748184163 | 21.7 | 0.957850823 | 49.9 |
| 9 | 0.1 | 0.1 | 1 | 20 | 7.910103352 | 2.1 | 4.388324832 | 21.9 |
| 6 | 0.001 | 0.1 | 1 | 20 | 0.554912387 | 40.7 | 0.383127444 | 58.9 |
| 6 | 0.01 | 0.1 | 1 | 20 | 0.965436275 | 31.9 | 0.569898229 | 56.1 |
| 6 | 0.1 | 0.1 | 1 | 20 | 1.748184163 | 21.7 | 0.957850823 | 49.9 |
| 6 | 1 | 0.1 | 1 | 20 | 4.614267919 | 7.5 | 2.318432829 | 38.9 |
| 6 | 0.1 | 0.01 | 1 | 20 | 1.748243716 | 21.7 | 0.957881043 | 49.9 |
| 6 | 0.1 | 0.1 | 1 | 20 | 1.748184163 | 21.7 | 0.957850823 | 49.9 |
| 6 | 0.1 | 1 | 1 | 20 | 1.748243716 | 21.7 | 0.957881043 | 49.9 |
| 6 | 0.1 | 10 | 1 | 20 | 1.742890133 | 22.0 | 0.956784532 | 49.7 |
| 6 | 0.1 | 0.1 | 0.1 | 20 | 0.454210326 | 42.9 | 0.313027213 | 60.7 |
| 6 | 0.1 | 0.1 | 0.5 | 20 | 1.692230119 | 21.2 | 0.932634623 | 50.8 |
| 6 | 0.1 | 0.1 | 1 | 20 | 1.748184163 | 21.7 | 0.957850823 | 49.9 |
| 6 | 0.1 | 0.1 | 2 | 20 | 1.938998043 | 19.4 | 1.040855977 | 49.0 |
| 6 | 0.1 | 0.1 | 1 | 5 | 0.503622804 | 41.1 | 0.335861445 | 60.8 |
| 6 | 0.1 | 0.1 | 1 | 10 | 1.830046162 | 20.4 | 1.005303425 | 48.6 |
| 6 | 0.1 | 0.1 | 1 | 20 | 1.748184163 | 21.7 | 0.957850823 | 49.9 |
| 6 | 0.1 | 0.1 | 1 | 50 | 1.824167202 | 21.4 | 0.998465185 | 49.3 |

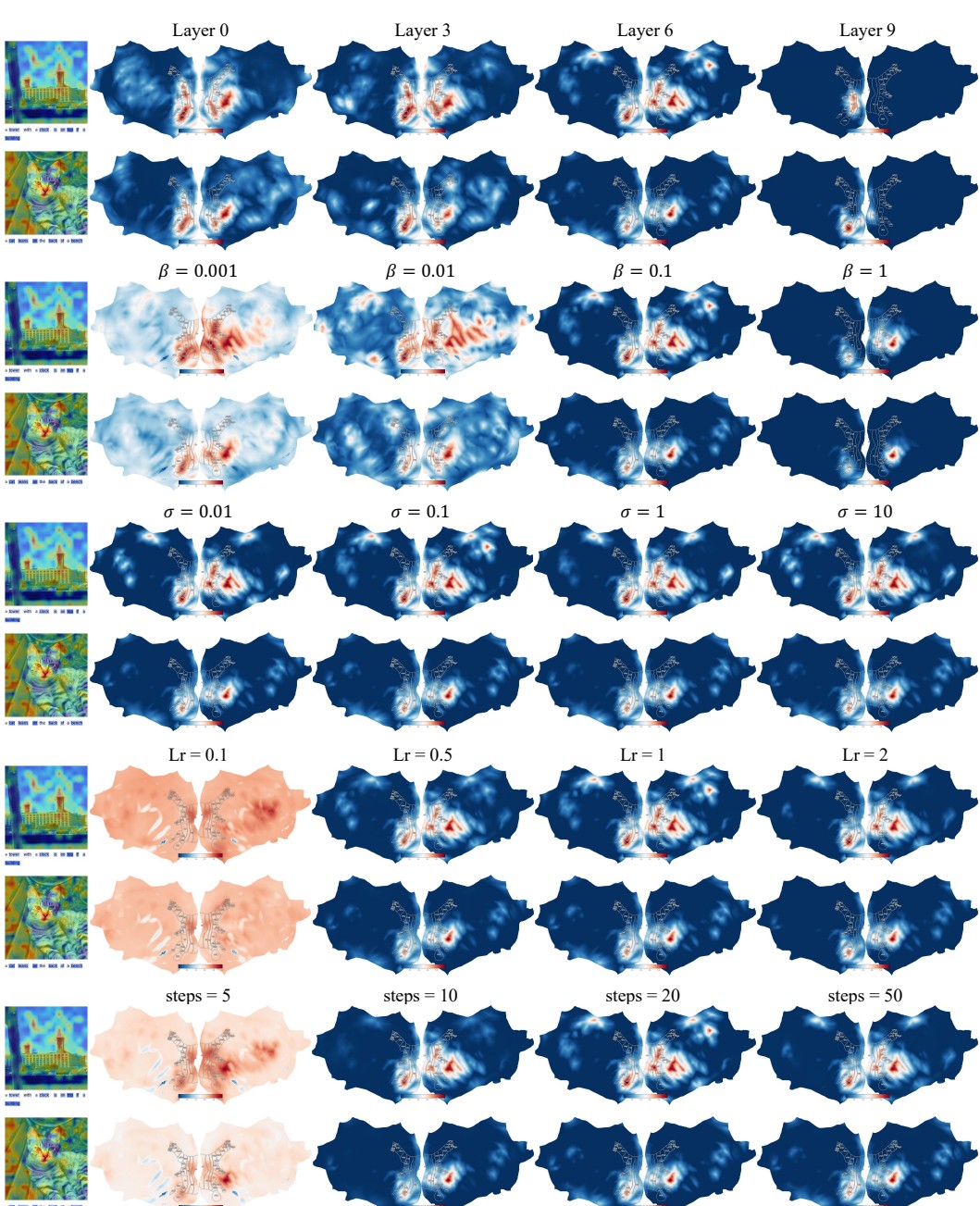

Figure 10: Hyperparameter sensitivity analysis of the IB-BA method. From top to bottom: target layer, $\beta$, variance of the Gaussian noise, learning rate, and training steps.

# D  EXTENDED TEXT DESCRIPTION

## D.1  COCO CAPTIONS RETRIEVAL

To illustrate the necessity of detailed text descriptions, we conducted the following cross-modal retrieval evaluation. We used three pretrained CLIP models (`CLIP-ViT-B/32`, `CLIP-ViT-L/14`, and `CLIP-ViT-H/14`) as vision-language models to extract the performance of COCO captions versus detailed descriptions generated by Llava1.6-13B and Qwen2.5-VL-32B in cross-modal retrieval tasks. As shown in Fig. 11, using detailed text descriptions significantly improves retrieval

performance, demonstrating their effectiveness in capturing richer semantic information for better

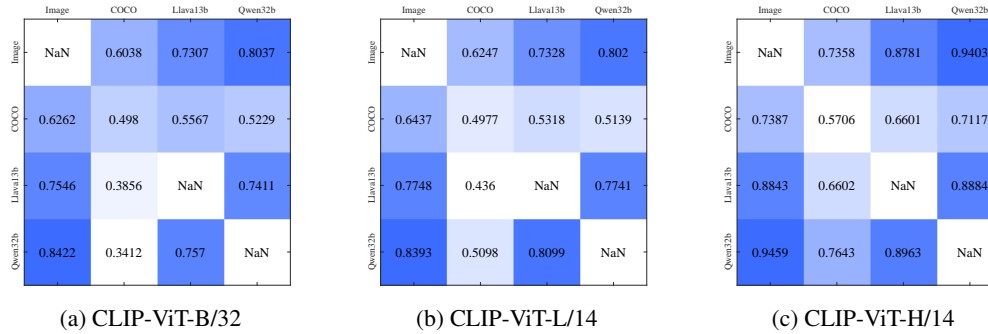

(a) CLIP-ViT-B/32     (b) CLIP-ViT-L/14     (c) CLIP-ViT-H/14

Figure 11: Cross-modal retrieval Top 1 results using COCO captions versus detailed descriptions generated by Llava1.6-13B and Qwen2.5-VL-32B.

## D.2 GENERATE DETAILED TEXT DESCRIPTIONS

```
<|im_start|>system
You are an AI visual assistant that can analyze objects in the
    image. Currently, you receive an image and some sentences
    each describing the image you are observing. \n Please
    describe objects and relevance, concepts, background, color
    and scene of the image in a detailed manner but without
    decoration and embellishment. \n Always answer as if you are
    directly looking at the image. \n Describe the image content
    clearly and concisely and retain the meanings of each
    objects, relevance, concepts, background, color and scene in
    the image. \n Describe the image directly from the
    beginning. Do not with \'The image shows\' or \'The image
    depicts\'. Don\'t summarize or overall. \n Keep your answer
    less than 77 characters and words.
<|im_end|>

<|im_start|>user
{NSD captions of the image from the COCO dataset}
<|vision_start|><|image_pad|><|vision_end|>
<|im_end|>

<|im_start|>assistant
...
```

## D.3 TEXT DESCRIPTIONS EXAMPLES FROM QWEN2.5-VL-32B

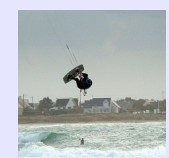 A person is kite surfing over ocean waves near the shore. They are airborne, performing a trick while holding onto the kite lines. The background includes houses and a clear sky. The water is choppy, indicating windy conditions.

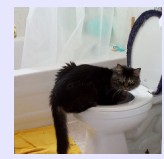 A black and gray cat is sitting on the edge of a white toilet, facing left toward an open bathroom door. The scene includes a yellow mat on the floor and a shower curtain in the background.

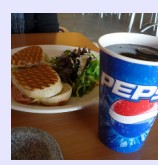

A hand uses a knife to cut a homemade pizza on a wooden board. Next to it, there's a bowl of salad with utensils, two glasses of white wine, and two wine bottles on a wooden table. The scene suggests a casual dining setup.

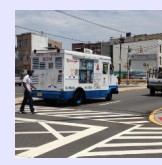

A wooden table holds a white plate with a grilled sandwich (possibly a panini) and a side of green salad. Next to it is a blue Pepsi cup filled with dark soda. The background includes part of a person's arm and a tiled floor. The setting appears casual, likely a café or dining area.

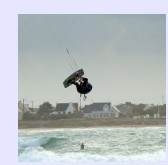

A man is crossing a street near an ice cream truck. The truck is white with colorful designs and text, parked on the right side of the road. The background includes urban buildings and a clear sky. The scene appears to be a busy intersection.

### D.4 TEXT DESCRIPTIONS EXAMPLES FROM LLAVA-V1.6-13B

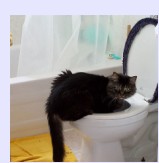

The image shows a person kiteboarding in the ocean on a windy day. The individual is holding onto a kite and a surfboard. The surfboard is yellow and black. In the background, there are buildings and cars. The sky is gray and cloudy.

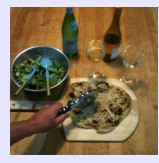

A black cat is sitting on the edge of a white toilet, looking toward the open bathroom door. The cat has a fluffy tail. The toilet is next to a bathtub with a blue shower curtain. The bathroom floor is covered with a yellow rug.

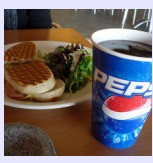

A hand holding a pizza cutter is slicing a pizza on a table. The table has a wooden cutting board, a green bottle, two glasses, a metal bowl with salad, and a wooden table. The hand is wearing a watch. The scene suggests a meal preparation.

The image displays a dining table with a white plate containing a grilled sandwich and a salad. Beside the plate is a blue cup with the Pepsi logo and ice cubes inside. The table is wooden and there's a person seated at the table. The background is blurred but suggests an indoor setting with additional furniture and chairs.

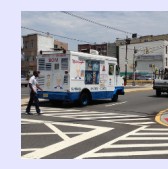

A man in white shirt walking across street. Blue and white truck with ice cream sign parked on street. Truck has window and door. Red traffic light. Buildings in background.

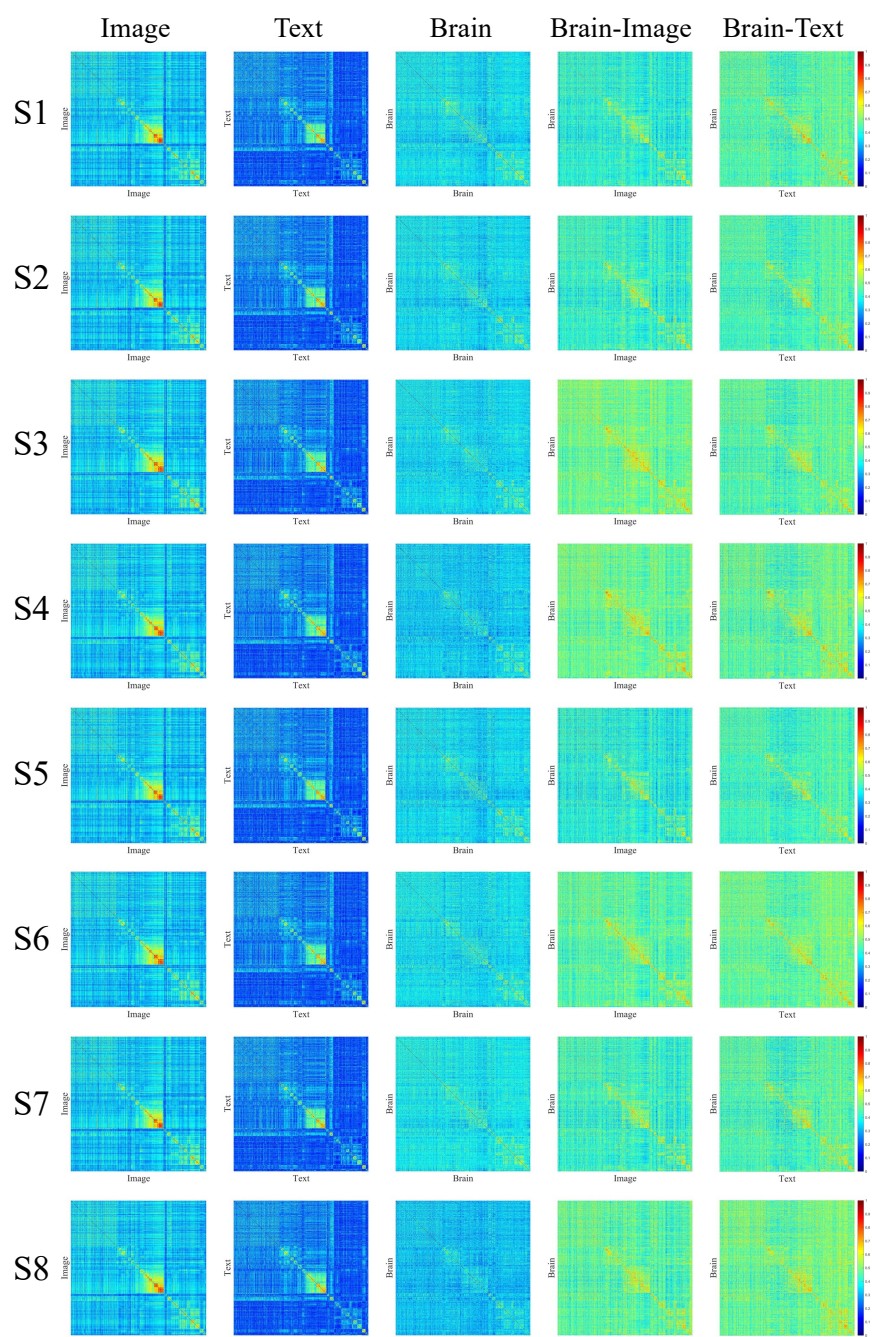

Figure 12: Representation similarity matrix (RSM) of brain visual-language embeddings.

