# OpenReview forum: "Bridging Vision, Language, and Brain: Whole-Brain Interpretation of Visual Representations via Information Bottleneck Attribution"
_ICLR.cc/2026/Conference — ICLR 2026 Conference Withdrawn Submission_

### Official Review · Reviewer_q7GV · 2025-10-26

**Soundness:** 2
**Presentation:** 3
**Contribution:** 2
**Rating:** 2
**Confidence:** 4

**Summary:**

This paper presents an Information Bottleneck approach to an fMRI decoding analysis based on NSD. The paper is an interesting read and it’s clear a lot of work went into it, but I think that this comes out of trying a few analyses and making good plots and then collecting this into a documentation of a series of experiments without much of a cohesive story that pushes the scientific frontier in terms of adding new knowledge and understanding. The figures don’t make sense to me and the captions do not help at all and are not what I would expect in a solid submission to this conference. There is a focus on providing nice-looking images and equations. Much of what takes up space in the paper makes more sense to be in the appendix and I think there is an organisational issue. I would have liked to see a cohesive and strong story to be developed, where the authors can then pull the relevant experiments together to tell that story. As it stands now, it seems like all the figures and experiments were bundled together and the question was asked, “How can we tell a story with what we have?” Decisions are made and not explained (e.g. why select only subject 01 for an ablation analysis - if there is no motivation and just because you had to pick one, then say that). Sometimes the authors talk about the deficiency of COCO captions so a fancy new method of image annotation is presented, but then this is dropped when approaching another analysis (without explanation). Overall, this is a good piece of work but it lacks the scientific analysis needed to pass the bar for ICLR, I think. I’m not left with a strong desire to tell a colleague about a finding here based just on the images shown without much explanation. I don’t feel that the contribution of the information bottleneck approach is sufficiently explained.

**Strengths:**

- An honest literature review highlighting opposing views, refreshing and more complete
- Inclusion of code in supplementary material was good (a quick review of it all looked good  to me)
- Engagement with prior models and good comparisons, although these are not readily contrasted and compared in the text
- The ablation study was interesting to read, but the conclusions were quite general and not easy to see beyond just applications to those specific images

**Weaknesses:**

- “The NSD dataset” is fine in speech, but in formal academic writing, the repetition of “dataset” looks odd. I’d say, “We evaluate the proposed methods on (the) NSD, …”
- Figure 1’s caption is not sufficient. What is the number next to the subjects? Is it the number of voxels identified? While I think this is the case, I don’t think I should be left to guess at this. Zooming in at 500% (the maximum I can do) I am not sure what I’m looking at fully. Are those the parcellation labels in white? This looks cluttered. I just feel that “high response to visual stimulation” with no other details to be wholly lacking and uninformative, especially when the appendix is so full of more interesting figures and graphs. What’s the threshold for determining “responsiveness” here. Is this still from NSD? Is it a noise ceiling that was used to calculate this? I hope you can see that the presence of this figure in its current form raises more questions than I think it’s supposed to clarify.
- Figure 2 is also quite weird. I think we can infer the grey area from the non-red voxels would be the entire cortex and this doesn’t need to be red. Also, when working with colour in figures, please address the usage of colours in the caption. The demarcations of what I think are the parcellation boundaries in the whole-cortex just look weird in that case. Additionally, “whole-brain cortex” is a non-standard term that sounds very funny to me. Whole-brain analyses are analyses that can extend in any part of the brain, deep in non-cortical structures, so you can’t call the cortex a “whole-brain” analysis. I’d rectify the usage of this terminology to be clearer.
- I have a problem with this sentence: “In contrast, our whole-brain model incorporates all voxels”. While I know that later you talk about using 3D fMRI images, e.g. an actual whole-brain analysis, the plots in Figures 1-3 are all of surfaces and mesh plots of vertices derived from the 3D space. I think a reader would be confused (as I was initially) to think you are working in surface space because that’s all that is plotted, to then see you actually are working with voxels in a whole-brain analysis. I think you need to point this out somewhere, either in captions or the text, that you’re looking at surface plots for visualisation of the method.
- I think you need to make more of a motivation as to why using non-visual regions automatically mean it’s better and more comprehensive to include them. We know a lot of the non-visual regions are not going to be driven by the stimulus that was the goal of collecting NSD (e.g. natural scenes) so what’s the direct logic to open up the analysis to all surface vertices?
- Some analyses have been done and appear to be included to show nice pictures, but are not tied to the original research question at all. Looking at Figure 12 and the RSMs (again with a caption that is no help to me) I am not sure what I should be getting out of these RSMs. What conclusion or observation do you want to support with these? Perhaps I missed something, but it feels like it’s presented because that analysis was done, not because it supports any scientific or experimental conclusions.
- Figures 5-6, please use informative captions. What are you trying to tell the reader with these plots? Please feel free to put the variational distribution equations (that take up a lot of space) into the appendix and that should free up some more room to be able to talk more about how the plots support analyses and conclusions you want to draw the reader to.

**Questions:**

1. “For attribution analysis with IB-BA, however, we revert to the original COCO captions” - why?
2. Evaluation metrics, retrieval accuracy, this isn’t fully explained. What domain is being used, the entire dataset, at a batch level? This is really important information that has been left out!
3. Can you provide a response that can tie together a more coherent theme for this submission? I want to know more than a new method was tried and here are the nice pictures. I see a lot of reporting of results and tables with numbers in, but there just isn't a coherent message behind it about what problem you're tackling. It's a huge one that no single paper could solve, but the end result being "better alignment" doesn't mean much when every 2nd paper using NSD says the same thing.

---

### Official Review · Reviewer_MMSc · 2025-10-27

**Soundness:** 2
**Presentation:** 2
**Contribution:** 2
**Rating:** 2
**Confidence:** 3

**Summary:**

Neuroscience and computational neural net models are still relatively far apart on the issue of multimodality, the authors argue.  They claim to bridge the gap by advocating a whole brain representation perspective (instead of representation of isolated voxels) and to more interpretable brain representations,  They use a bottleneck approach an measuring mutual information between conceptual and visual representations and brain activity.

**Strengths:**

These are exciting topics for sure and the full brain approach developed in this paper seems to do much better on alignment issues as the empirical work shows.

**Weaknesses:**

In general this paper seems to have two topics instead of one : brain multimodal representation alignment,
and brain attribution.  Cramming two complex topics into one paper leads to a number of unanswered questions (below) and a certain lack of coherence, as at least I didn't see how the two topics really came together.  It wasn't even completely clear to me what the core question for brain attribution was, though I think I guessed correctly (see below).  That said, however, I think that the alignment results are interesting and when developed further would warrant acceptance.

The main cognitive question, how the human brain integrates visual and linguistic information, is not really treated in much detail.   It would be super interesting if the authors addressed this question in more detail, though I admit I am not an expert in this area.  The brain attribution method is supposed to interpret relationships between conceptual and visual representations and brain activity but the details aren't there.  I would strongly suggest separating this paper into two, one a basic study laying out the issues and maybe concentrating on the clip to brain alignment and a second, follow up study with more detailed experiments laying out how the bottleneck viewpoint helps  us understand better brain areas related to visual and conceptual or linguistic representation (I take it that's the attribution question).

Some specifics:
1) Why talk about model attribution in the related work?  You're interested in brain modeling.   This section seems to indicate that you're going to talk about computational model interpretability, which you really don't do.

2) Doesn't the full brain model become harder to interpret than the alternatives which provide an a priori limit to the cortical regions studied?  You have way more voxels.   So in principle way more elements to put in the right places.  What in fact are you aligning?  Whole brain activity or particular patterns in particular areas?

3) You say your method preserves brain geometry (spatial structure) but your approach takes a curved 3d object (manifold) and "flattens it".  Flattening suggests a projection from a curved manifold to a 2d or at least a Euclidean space.  Flattening, as you can see from a Mercator map projection on the globe, introduces distortions and only locally preserves geometry.   Is the flattening really what you're doing?

4) While the Clip alignment of brain embeddings with visual and language sources is clear, are the experiments on alignment also using the information bottleneck principle?  From the section on Model Implementation, it looks like the  figures in Figure 5 are just based on CLIP.  If so please say that.  If not please explain how the information bottleneck is being used there.

5) Also as many people do, you make assumptions about distributions that may or may not hold up empirically.  This weakens the analysis from a stanpoint of interpretability.  Is there good reason to assume such Gaussian distributions, given that $p(Z_B)$ is intractable?  What do we know about those distributions?

6) In general I did not see a clear connection between the alignment and the brain attribution sections
It would help to go into more detail about the tasks for non specialist readers.   More details would help but of course would extend the paper perhaps beyond the page limit, whence my suggestion that you make 2 papers out of this submission.

**Questions:**

Please see the questions above

---

### Official Review · Reviewer_NyHL · 2025-10-31

**Soundness:** 3
**Presentation:** 3
**Contribution:** 2
**Rating:** 4
**Confidence:** 4

**Summary:**

This paper proposes a whole-brain representation module and an Information Bottleneck-based Brain Attribution (IB-BA) method to bridge fMRI brain activity with visual and linguistic modalities. By aligning brain embeddings with pre-trained CLIP image/text embeddings via contrastive learning and extending information-theoretic attribution to a tri-modal setting, the work achieves state-of-the-art cross-modal retrieval performance and more interpretable cortical attribution maps on the NSD dataset. The research addresses a meaningful gap between cognitive neuroscience and AI, but its novelty is partially constrained by reliance on existing contrastive learning and information bottleneck frameworks, and some experimental analyses lack sufficient depth.

**Strengths:**

1. The whole-brain representation module effectively preserves voxel spatial topology and captures distributed brain dynamics, addressing the limitation of existing methods that focus on isolated voxels or partial brain regions.

2. IB-BA extends information-theoretic attribution to a tri-modal (brain-image-text) setting, providing a principled approach to identify informative voxels without relying on explicit ground-truth labels for neural data.

3. Extensive experiments on the large-scale NSD dataset demonstrate superior cross-modal retrieval performance and more interpretable attribution maps compared to baseline methods, with solid ablation studies on key components (modality alignment, brain regions, caption sources).

**Weaknesses:**

1. The core technical frameworks (contrastive learning for alignment, information bottleneck for attribution) are built on existing methods (CLIP, IBA, M2IB), with the main innovation lying in adaptation to tri-modal brain data rather than fundamental methodological breakthroughs.

2. The analysis of IB-BA’s hyperparameter sensitivity (e.g., target layer, β) is descriptive but lacks quantitative justification for the chosen parameters (e.g., why β=0.1 is optimal beyond "balanced trade-off").

3. The paper does not discuss the computational complexity of the whole-brain encoder and IB-BA, which is critical for scalability to larger datasets or real-time applications.

4. Individual differences among the 8 NSD subjects are observed in results (e.g., Table 4, Figure 8) but not analyzed—why some subjects yield significantly higher retrieval accuracy remains unexplained.

5. The use of LLM-generated captions (Qwen2.5-VL-32B) improves alignment but introduces potential biases; the paper does not validate whether these generated descriptions accurately reflect the original stimuli or affect attribution reliability.

**Questions:**

1. What is the computational overhead of the proposed whole-brain representation module compared to baseline brain decoding models (e.g., Ridge, Mindeye)?

2. Can the authors provide a more rigorous justification for the choice of β=0.1 in IB-BA, such as ablation across more β values or theoretical bounds on the trade-off between informativeness and compression?

3. How do individual differences in brain structure/function (e.g., voxel response variability shown in Figure 1) affect the generalizability of the proposed model to unseen subjects?

4. Are there any systematic biases in the LLM-generated captions, and how do these biases impact the alignment between brain and text embeddings?

---

### Official Review · Reviewer_eFep · 2025-11-01

**Soundness:** 3
**Presentation:** 3
**Contribution:** 3
**Rating:** 6
**Confidence:** 3

**Summary:**

This paper tackles the challenge of understanding how the human brain integrates visual and linguistic information by proposing two main contributions. First, the authors develop a whole-brain representation module that aligns fMRI signals with CLIP image and text embeddings through contrastive learning, utilizing 3D patch embeddings and transformer-based architectures to preserve spatial topology and capture distributed brain dynamics. Second, they introduce Information Bottleneck-based Brain Attribution (IB-BA), extending information-theoretic attribution to a tri-modal (brain-image-text) setting to identify informative voxel subsets. Experiments on the Natural Scenes Dataset demonstrate improved cross-modal retrieval performance and interpretable cortical attribution maps compared to existing methods.

**Strengths:**

1. Prior work's focus on pre-selected voxels is genuinely problematic; whole-brain modeling is a meaningful advance
2. The combination of alignment and attribution provides both predictive performance and interpretability
3. 8 subjects from NSD, comparison against multiple baselines.

**Weaknesses:**

1. The paper is primarily an engineering contribution combining existing techniques. The tri-modal extension of IB is straightforward—essentially adding another term to the fitting objective.
2. While Figure 7 and Table 6 show sensitivity analyses, the paper doesn't provide clear guidelines for selecting \beta, target layer, etc. for new applications. The choice seems fairly arbitrary.

**Questions:**

1. Can a model trained on one subject decode brain activity from another?
2. How tight are the variational bounds in Equations 5-7? Can you empirically estimate the gap between the true and approximated objectives?

---

### Note · Authors · 2025-12-11

I have read and agree with the venue's withdrawal policy on behalf of myself and my co-authors.